# Effects of intrathecal administration of sodium nitroprusside and nicardipine on cerebral pial microcirculation, cortical tissue oxygenation, and electrocortical activity in the early post-resuscitation period in a porcine cardiac arrest model

Hyoung Youn Lee[1], Najmiddin Mamadjonov[2], Kyung Woon Jeung[3,4]*, Yong Hun Jung[3,4], Wan Young Heo[3], Yong Soo Cho[3,4], Byung Kook Lee[3,4], Tag Heo[3,4]

1 Trauma Center, Chonnam National University Hospital, Gwangju, Republic of Korea, 2 Department of Medical Science, Chonnam National University Graduate School, Gwangju, Republic of Korea, 3 Department of Emergency Medicine, Chonnam National University Hospital, Gwangju, Republic of Korea, 4 Department of Emergency Medicine, Chonnam National University Medical School, Gwangju, Republic of Korea

* neoneti@hanmail.net

## Abstract

Recent studies suggested intrathecal vasodilator administration as a therapy to mitigate post-ischemic cerebral hypoperfusion following cardiac arrest. We examined the effects of two commonly used intrathecal vasodilators, sodium nitroprusside (SNP) and nicardipine, on cerebral pial microcirculation, cortical tissue oxygen tension ($PctO_2$), and electrocortical activity in the early post-resuscitation period using a porcine model of cardiac arrest. Thirty pigs were resuscitated after 14 min of untreated cardiac arrest. At 30 min after resuscitation from cardiac arrest, 30 pigs randomly received 4 mg of SNP, 4 mg of nicardipine, or saline placebo via subdural intracranial catheters and were observed for 5 h. Group effect and group-time interaction were assessed using linear mixed-effects models. The mean arterial pressure was lower in the nicardipine group (coefficient [95% confidence interval {CI}], -15.824 [-24.082 to -7.566]) and higher in the SNP group (coefficient [95%CI], 11.232 [2.974–19.490]) compared to the saline group. The percentage of pial arteriole diameter relative to the pre-arrest baseline value (coefficient [95% CI], 48.970 [13.884–84.057]), microvascular flow index (coefficient [95% CI], 0.296 [0.071–0.521]), and $PctO_2$ (coefficient [95% CI], 26.926 [12.404–41.449]) were higher in the SNP group but not in the nicardipine group compared to the saline group. Amplitude-integrated electroencephalography amplitude recovery was faster in the SNP group (coefficient [95% CI], 1.149 [0.468–1.829]) but not in the nicardipine group compared to the saline group. In conclusion, intrathecal SNP but not nicardipine was effective in treating post-ischemic cerebral hypoperfusion after cardiac arrest.

**Data Availability Statement:** All relevant data are within the manuscript and its Supporting Information files.

**Funding:** This work was supported by the National Research Foundation of Korea (NRF) grant (NRF-2021R1A2C1003390 and NRF-2022R1A2C1012733) funded by the Korean government (MSIT) and a grant (BCRI24034) of Chonnam National University Hospital Biomedical Research Institute. The funders had no role in the study design, data collection and analysis, decision to publish, or preparation of the manuscript.

**Competing interests:** The authors have declared that no competing interests exist.

## Introduction

Despite the availability of current therapies for mitigating hypoxic-ischemic brain injury (HIBI) after cardiac arrest, the majority of patients resuscitated from cardiac arrest suffer from severe neurologic sequalae or die due to HIBI [1,2]. Therefore, additional effective treatments for mitigating HIBI following cardiac arrest are needed to reduce the morbidity and mortality of these patients.

Multiple studies have suggested post-ischemic cerebral hypoperfusion, a reduction in cerebral blood flow (CBF) occurring early after resuscitation from cardiac arrest, as an important therapeutic target for reducing HIBI [3–7]. Although various therapies have been proposed to mitigate post-ischemic cerebral hypoperfusion [5–7], none have been widely accepted in clinical practice. Intrathecal vasodilator administration has been used as a rescue therapy for patients with treatment-refractory cerebral vasospasm after subarachnoid hemorrhage [8–11]. Two recent studies demonstrated the potential of intrathecal vasodilator administration in mitigating post-ischemic cerebral hypoperfusion following cardiac arrest [12,13]. In a study using a porcine cardiac arrest model [12], administration of sodium nitroprusside (SNP) into the subarachnoid space effectively mitigated cerebral cortical hypoperfusion, reduced the duration of exposure to cerebral cortical tissue hypoxia, and promoted the recovery of electrocortical activity. In another study using minipigs in a cardiac arrest model [13], CBF improvement by SNP administration led to a decrease in cerebral lactate. As only SNP was tested in these studies [12,13], it is necessary to identify optimal intrathecal vasodilators for treating post-ischemic cerebral hypoperfusion following cardiac arrest.

In this study, to identify optimal intrathecal vasodilators, we examined the effects of SNP and nicardipine, two intrathecal vasodilators commonly used to reverse cerebral vasospasm in patients with subarachnoid hemorrhage [10,11,14–23], on cerebral pial microcirculation, cortical tissue oxygenation, and electrocortical activity in the early post-resuscitation period using a porcine cardiac arrest model. We hypothesized that intrathecal administration of SNP and nicardipine would have similar effectiveness in improving pial microcirculation, cortical tissue oxygenation, and electrocortical activity.

## Materials and methods

This prospective experimental study was conducted using 36 Yorkshire/Landrace cross pigs of either sex weighing 22.8 ± 1.2 kg. All experiments were approved by the Animal Care and Use Committee of Chonnam National University Hospital (CNUH IACUC-22028) and performed in accordance with the National Institutes of Health Guide for the Care and Use of Laboratory Animals. The investigators who performed the experiments had completed an Institutional Animal Care and Use Committee training course on animal care and handling. All surgical interventions were performed under sevoflurane anesthesia, and every effort was made to prevent unnecessary suffering of the animals.

### Animal preparation

The animals were acclimated in a light- and temperature-controlled room (12 h light/dark cycle; 21°C) with ad libitum access to commercial feed and tap water for 7 days before the experiment. The animal preparation procedures have been previously described in detail [12]. After intramuscular administration of ketamine (20 mg/kg) and xylazine (2.2 mg/kg), they were anesthetized with sevoflurane, endotracheally intubated, and mechanically ventilated with a mixture of nitrous oxide and oxygen (7:3). Sevoflurane was titrated to achieve an appropriate anesthetic depth (no pedal withdrawal reflex and no increase in arterial pressure or heart rate). End-tidal carbon dioxide (ETCO$_2$) was monitored, and the ventilation rate was

adjusted to maintain ETCO$_2$ within the normal physiologic range. Rectal temperature was monitored and maintained at 37–38˚C using a heating blanket. The right femoral artery was cannulated with a 7-F catheter for arterial pressure measurement and blood sampling. The right external jugular vein was cannulated with a 7-F catheter for 0.9% sodium chloride solution (5 mL/kg/h) and drug administration. After placing the animals in the prone position, L2 laminectomy was performed to expose the dura, through which a 5-F catheter was inserted into the subarachnoid space for cerebrospinal fluid (CSF) pressure measurement. A scalp incision was made to expose the skull, and four cranial burr holes (1 cm in diameter) were created over the left and right parietal cortices. Two burr holes were located 1 cm posterior to the coronal suture and used to assess pial microcirculation and cortical tissue oxygenation. A closed cranial window for assessment of pial microcirculation was created on one burr hole as described previously [12]. An optical oxygen sensor (PM-PSt7; PreSens-Precision Sensing GmbH, Regensburg, Germany) was introduced through the other burr hole, and its tip was placed on the cerebral cortex for cortical tissue oxygen tension (PctO$_2$) measurement. The remaining two burr holes were located over the right and left parietal cortices 3 cm posterior to the coronal suture; through each burr hole, a 20 G catheter (Perifix ONE® Pediatric Epidural Anesthesia Catheter; B. Braun, Bethlehem, PA) was introduced into the subarachnoid space for intrathecal administration. To monitor the amplitude-integrated electroencephalography (aEEG) signal, two needle electrodes were inserted into the left and right parietal regions, corresponding to the P3 and P4 positions of the international 10–20 system, respectively, and a reference electrode was inserted into the frontal region.

## Experimental protocol

A subcutaneous pocket extending approximately 3 cm under the xiphoid process was created following a skin incision immediately below the xiphoid process. After obtaining baseline measurements, ventricular fibrillation was induced with a brief application of an alternating current (60 Hz, 30 mA) through a pacing catheter placed near the right ventricle through the pocket (Fig 1). After 14 min of untreated cardiac arrest, cardiopulmonary resuscitation was started with mechanical chest compressions (Life-Stat; Michigan Instruments, Grand Rapids, MI; depth, 20% of the anteroposterior chest diameter; frequency, 100 compressions/min),

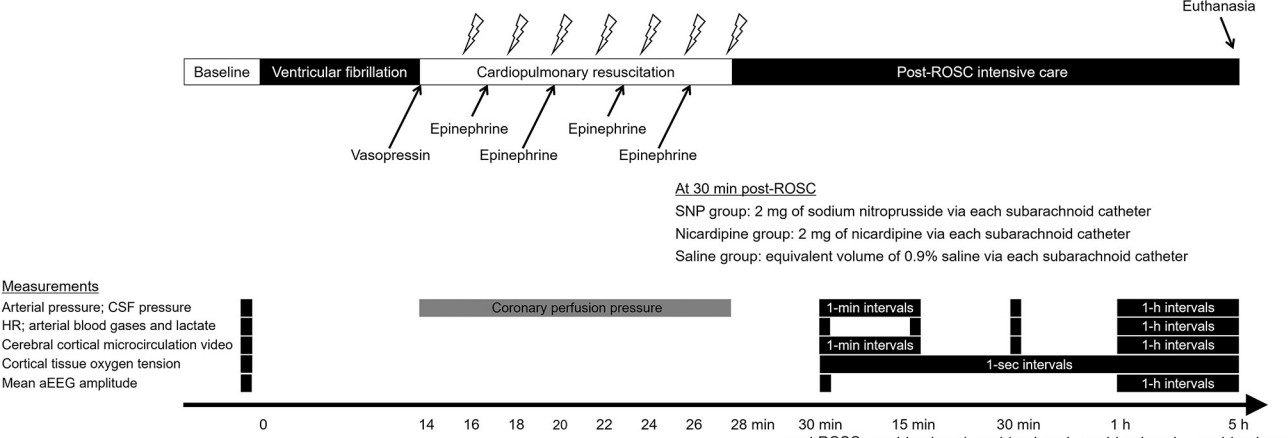

**Fig 1. Experimental timeline.** The lightning marks indicate the onset of a pause of 10 s in chest compression for rhythm analysis and a biphasic 200 J electric countershock, if indicated. ROSC, return of spontaneous circulation; SNP, sodium nitroprusside; CSF, cerebrospinal fluid; HR, heart rate; aEEG, amplitude-integrated electroencephalography.

manual ventilations (frequency, 10 ventilations/min) with high-flow oxygen (15 L/min), and intravenous vasopressin injection (0.4 U/kg). During cardiopulmonary resuscitation, electrical defibrillation was attempted with a single biphasic 200 J countershock every 2 min. If return of spontaneous circulation (ROSC) was not achieved with the first electrical defibrillation, 0.02 mg/kg epinephrine was administered intravenously every 3 min. If ROSC could not be obtained within 14 min, resuscitation efforts were terminated for futility.

Animals that achieved ROSC received 5.5 h of intensive care under sevoflurane anesthesia. Immediately after ROSC, mechanical ventilation was restarted with a fraction of inspired oxygen at 100%. Fifteen minutes after ROSC, the ventilation rate and fraction of inspired oxygen were adjusted to maintain an $ETCO_2$ of 35–40 mmHg and a peripheral oxygen saturation of 94–98%, respectively. Norepinephrine was infused to maintain a mean arterial pressure (MAP) of at least 65 mmHg. An investigator randomly assigned the animals to the saline, SNP, or nicardipine group based on the information in a sealed envelope and prepared a saline, SNP (4 mg), or nicardipine (4 mg) solution in an equal volume (4 mL). The doses of SNP and nicardipine were chosen based on clinical studies in which intrathecal administration of SNP or nicardipine was used to prevent or treat cerebral vasospasm following subarachnoid hemorrhage [10,14,23]. All other investigators were kept blinded to group assignments during the experiment. The assigned drug was administered through the catheters introduced into the subarachnoid space (2 mL through each catheter) at 30 min after ROSC. None of the animals reached the predetermined humane endpoints for euthanasia (systolic arterial pressure <60 mmHg, heart rate <40 beats/min, or seizure) until the end of the post-ROSC intensive care period. Immediately after the post-ROSC intensive care period, the animals were humanely euthanized with a rapid intravenous bolus injection of potassium chloride under deep sevoflurane anesthesia (≥5%).

## Measurements

The primary outcomes were pial microcirculation parameters and $PctO_2$. Pial microcirculation videos were obtained using a hand-held digital microscope (G-Scope G5; Genie Tech, Seoul, Korea) placed over the closed cranial window at the pre-arrest baseline, immediately prior to the assigned treatment, at 1 min intervals for 15 min after the assigned treatment, at 30 min and 1 h after the assigned treatment, and every 60 min thereafter. The microscope (×250 magnification) showed an area of interest of $1,800 \times 1,000 \ \mu m^2$. Two investigators blinded to group assignments independently reviewed the obtained videos and determined the microvascular flow index (MFI; normal flow = 3, sluggish flow = 2, intermittent flow = 1, and absent flow = 0) [24]. In addition, a pial arteriole was selected in the video obtained at the pre-arrest baseline, and the diameter of the arteriole at each time point was measured at the same location using ImageJ (Softonic; National Institute of Mental Health, Bethesda, MD). To account for inter-animal variations, the arteriole diameter was expressed as a percentage of the pre-arrest baseline value (%arteriole diameter). The measurements from the two investigators were averaged for statistical analysis. $PctO_2$ was sampled using an oxygen meter (OXY-1 ST; PreSens-Precision Sensing GmbH, Regensburg, Germany) at the pre-arrest baseline, immediately prior to the assigned treatment, and at 1 s intervals after the assigned treatment. Arterial pressure and CSF pressure were sampled at the pre-arrest baseline, immediately prior to the assigned treatment, at 1 min intervals for 15 min after the assigned treatment, at 30 min and 1 h after the assigned treatment, and every 60 min thereafter. Heart rate, arterial blood gases, and arterial lactate levels were measured at the pre-arrest baseline, immediately prior to the assigned treatment, at 15, 30, and 60 min after the assigned treatment, and every 60 min thereafter. The aEEG signal was recorded using an electroencephalography machine (EEG-1250;

Nihon Kohden, Tokyo, Japan). The mean aEEG amplitude, which is the average of the voltage levels of the lower and upper margins of the aEEG band, was recorded at the pre-arrest baseline, immediately prior to the assigned treatment, and at 1 h intervals after the assigned treatment.

## Statistical analysis

We calculated the sample size for this study based on the MFI (mean ± standard deviation [SD], 2.14 ± 0.65; variance, 0.42) and %arteriole diameter (mean ± SD, 120.2 ± 38.1%; variance, 1455.3) from a pilot study. In the pilot study, the MFIs were 1.82 ± 0.47, 2.52 ± 0.63, and 2.09 ± 0.64 in the saline, SNP, and nicardipine groups, respectively, and the calculated within-group variance of the MFI was 0.34. The %arteriole diameters were 94.5 ± 21.1%, 125.7 ± 37.5%, and 140.2 ± 38.2% in the saline, SNP, and nicardipine groups, respectively, and the calculated within-group variance of the %arteriole diameter was 1,092.6. We calculated that 10 animals per group would be necessary to achieve an α of 0.05 and a power of 80% and included a total of 36 animals to minimize any effect of data loss. Statistical analyses were performed using R version 4.2.2 (R Foundation for Statistical Computing, Vienna, Austria) and T&F program version 4.0 (YooJin BioSoft, Goyang, Korea). The intraclass correlation coefficient was used to evaluate the interrater reliability of the MFI and %arteriole diameter measurements from the two investigators. Continuous data are reported as means ± SDs or medians with interquartile ranges, whereas categorical data are presented as counts and proportions. Comparisons among the three groups were conducted using one-way analysis of variance or the Kruskal-Wallis test for continuous variables and Fisher's exact test for categorical variables, followed by pairwise comparison with Bonferroni's adjustment. Linear mixed-effects models were constructed to analyze the time effect, group effect, and group-time interaction in serially obtained measurements after controlling for random effects derived from the repeated measurements. For each variable, the entire 5 h period after randomization was divided into subperiods according to the increase or decrease of data over time, and a linear mixed-effects model was constructed for each subperiod. If the inclusion of group-time interaction in a model caused severe multicollinearity, group-time interaction was excluded from the model. A two-tailed P value less than 0.05 was considered significant for all analyses.

## Results

Of the 36 animals used in this study, 6 animals failed to achieve ROSC and were excluded from the analysis. The remaining 30 animals were successfully randomized and survived throughout the post-ROSC intensive care period. There were no significant differences among the three groups in the pre-arrest baseline measurements, except for the MAP (Table 1). Although the pre-arrest baseline MAP differed among the three groups, it was within normal limits in all animals. Cardiopulmonary resuscitation variables and measurements immediately prior to the assigned treatment were comparable among the three groups (Table 2).

Fig 2 shows the hemodynamic parameters and arterial blood gases after the assigned treatment. Nicardipine induced a decrease in the MAP that lasted for around 30 min, whereas SNP induced a transient small decrease in the MAP immediately after administration, followed by a paradoxical increase in the MAP that lasted for 3 h. The mixed-effects model for the MAP during the first 5 min showed a significant group-time interaction among the three groups (P <0.001). The model revealed a significant decrease over time in the nicardipine group compared to the saline group (coefficient [95% confidence interval {CI}], -6.580 [-8.675 to -4.485]). The mixed-effects model for the MAP from 5 min to 5 h showed a significant group effect (P <0.001) and group-time interaction (P <0.001). The model revealed a lower MAP in the

**Table 1. Pre-arrest baseline measurements.**

| Variable | Saline group (N = 10) | SNP group (N = 10) | Nicardipine group (N = 10) | P value[a] |
|---|---|---|---|---|
| Weight (kg) | 22.4 ± 0.9 | 23.0 ± 1.5 | 22.9 ± 1.2 | 0.500 |
| Systolic arterial pressure (mmHg) | 122 (113–136) | 113 (107–121) | 113 (110–115) | 0.161 |
| Diastolic arterial pressure (mmHg) | 85 (76–95) | 71 (64–86) | 73 (68–75) | 0.065 |
| MAP (mmHg) | 103 (91–110) | 87 (79–101) | 88 (85–91) | 0.049 |
| Systolic right atrial pressure (mmHg) | 9 ± 2 | 9 ± 2 | 10 ± 1 | 0.471 |
| Diastolic right atrial pressure (mmHg) | 6 (5–7) | 5 (5–7) | 6 (5–7) | 0.880 |
| Mean right atrial pressure (mmHg) | 8 (6–9) | 7 (6–8) | 8 (6–8) | 0.825 |
| Heart rate (beats/min) | 96 (92–101) | 88 (86–108) | 86 (81–93) | 0.189 |
| Rectal temperature (˚C) | 37.3 ± 0.9 | 37.7 ± 0.6 | 37.8 ± 0.6 | 0.255 |
| Arterial pH | 7.545 ± 0.034 | 7.523 ± 0.032 | 7.512 ± 0.029 | 0.080 |
| $PaCO_2$ (mmHg) | 40 ± 3 | 40 ± 2 | 42 ± 3 | 0.074 |
| $PaO_2$ (mmHg) | 146 ± 21 | 137 ± 32 | 134 ± 32 | 0.620 |
| Arterial lactate (mmol/L) | 1.1 ± 0.4 | 1.2 ± 0.9 | 1.5 ± 0.5 | 0.343 |
| $PctO_2$ (mmHg) | 44.6 ± 12.1 | 38.3 ± 9.3 | 40.2 ± 9.4 | 0.392 |
| CSF pressure (mmHg) | 8 ± 2 | 9 ± 1 | 9 ± 2 | 0.216 |
| Mean aEEG amplitude (μV) | 43 ± 9 | 48 ± 8 | 42 ± 6 | 0.263 |
| MFI[b] | 3.0 | 3.0 | 3.0 | NA |
| Diameter of measured arteriole (μm) | 33 (29–75) | 45 (41–52) | 49 (30–75) | 0.798 |

Data are presented as mean ± SD or median with interquartile range. [a] P values were computed using one-way ANOVA or Kruskal-Wallis test for differences among the three groups. [b] The MFI at the pre-arrest baseline was 3 for all animals. SNP, sodium nitroprusside; MAP, mean arterial pressure; $PaCO_2$, partial pressure of arterial carbon dioxide; $PaO_2$, partial pressure of arterial oxygen; $PctO_2$, cortical tissue oxygen tension; CSF, cerebrospinal fluid; aEEG, amplitude-integrated electroencephalography; MFI, microvascular flow index.

nicardipine group (coefficient [95% CI], -15.824 [-24.082 to -7.566]) and a higher MAP in the SNP group (coefficient [95%CI], 11.232 [2.974–19.490]) than in the saline group. The dose (0.70 ± 0.69 mg, 0.30 ± 0.22 mg, and 1.29 ± 0.82 mg in the saline, SNP, and nicardipine groups, respectively) and duration (66 [39–144] min, 33 [24–55] min, and 98 [74–134] min in the saline, SNP, and nicardipine groups, respectively) of norepinephrine infusion after the assigned treatment differed significantly among the three groups ($P$ = 0.006 and 0.032, respectively). Post-hoc pairwise comparisons revealed a significantly higher dose ($P$ = 0.005) and a significantly longer duration ($P$ = 0.030) of norepinephrine infusion in the nicardipine group than in the SNP group. A small rapid increase in CSF pressure was observed immediately after intrathecal vasodilator administration in all groups. There was a significant group-time interaction with respect to CSF pressure in both the first 5 min period ($P$ = 0.020) and the period from 5 min to 5 h ($P$ = 0.041); however, no significant group effect was observed in either period. The mixed-effects model for $PaO_2$ during the first 1 h showed a significant group effect ($P$ <0.001) and group-time interaction ($P$ = 0.038). The model revealed a lower $PaO_2$ in both the nicardipine (coefficient [95% CI], -29.880 (-47.004 to -12.756)) and SNP (coefficient [95% CI], -31.100 [-48.224 to -13.976]) groups compared to the saline group. However, neither group effect nor group-time interaction was observed with respect to $PaO_2$ in the period from 1 h to 5 h. There was no significant group effect or group-time interaction in any period with respect to heart rate, $PaCO_2$, and arterial lactate.

Fig 3 shows the %arteriole diameter and MFI. The intraclass correlation coefficients of the MFI and %arteriole diameter were 0.869 (95% CI, 0.850–0.886) and 0.951 (95% CI, 0.937–0.961), respectively. In the saline group, the arteriole diameter was decreased to approximately

**Table 2. Cardiopulmonary resuscitation variables and measurements immediately prior to the assigned treatment.**

| Variable | Saline group (N = 10) | SNP group (N = 10) | Nicardipine group (N = 10) | P value[a] |
|---|---|---|---|---|
| Cardiopulmonary resuscitation variables | | | | |
| Number of epinephrine doses (N) | 1 (1–2) | 1 (1–1) | 1 (1–1) | 0.377 |
| Duration of cardiopulmonary resuscitation (min) | 4 (4–6) | 4 (4–4) | 4 (4–6) | 0.592 |
| Measurements immediately prior to the assigned treatment | | | | |
| Systolic arterial pressure (mmHg) | 111 ± 15 | 113 ± 11 | 113 ± 13 | 0.945 |
| Diastolic arterial pressure (mmHg) | 80 ± 16 | 75 ± 12 | 76 ± 13 | 0.646 |
| MAP (mmHg) | 92 ± 15 | 89 ± 11 | 91 ± 12 | 0.866 |
| Systolic right atrial pressure (mmHg) | 14 ± 3 | 13 ± 2 | 12 ± 2 | 0.132 |
| Diastolic right atrial pressure (mmHg) | 8 ± 3 | 8 ± 2 | 8 ± 2 | 0.824 |
| Mean right atrial pressure (mmHg) | 11 (9–12) | 11 (10–11) | 10 (8–11) | 0.422 |
| Heart rate (beats/min) | 168 ± 14 | 155 ± 22 | 154 ± 20 | 0.209 |
| Rectal temperature (˚C) | 36.9 ± 0.8 | 37.1 ± 0.6 | 37.2 ± 0.7 | 0.587 |
| Arterial pH | 7.170 (7.090–7.208) | 7.145 (7.080–7.158) | 7.145 (7.123–7.158) | 0.510 |
| PaCO$_2$ (mmHg) | 56 ± 7 | 59 ± 7 | 58 ± 10 | 0.652 |
| PaO$_2$ (mmHg) | 107 ± 21 | 89 ± 14 | 105 ± 21 | 0.077 |
| Arterial lactate (mmol/L) | 9.2 ± 2.6 | 8.6 ± 1.3 | 9.0 ± 1.2 | 0.785 |
| PctO$_2$ (mmHg) | 49.0 ± 17.6 | 43.1 ± 10.9 | 36.0 ± 13.8 | 0.148 |
| CSF pressure (mmHg) | 11 ± 2 | 11 ± 2 | 10 ± 2 | 0.542 |
| Mean aEEG amplitude (μV) | 10 ± 3 | 10 ± 3 | 10 ± 1 | 0.824 |
| MFI | 2.1 ± 0.6 | 2.3 ± 0.5 | 2.4 ± 0.7 | 0.225 |
| %Arteriole diameter (%) | 103 ± 25 | 118 ± 38 | 107 ± 25 | 0.585 |

Data are presented as mean ± SD or median with interquartile range. [a] P values were computed using one-way ANOVA or Kruskal-Wallis test for differences among the three groups. SNP, sodium nitroprusside; MAP, mean arterial pressure; PaCO$_2$, partial pressure of arterial carbon dioxide; PaO$_2$, partial pressure of arterial oxygen; PctO$_2$, cortical tissue oxygen tension; CSF, cerebrospinal fluid; aEEG, amplitude-integrated electroencephalography; MFI, microvascular flow index.

88% of the pre-arrest baseline value at 8 min after the assigned treatment (38 min after ROSC) and increased gradually thereafter. In this group, the MFI was decreased to approximately 42% of the pre-arrest baseline value at 5 min after the assigned treatment (35 min after ROSC). Although the MFI was gradually increased thereafter, it remained at levels below the pre-arrest baseline value until 3 h after the assigned treatment. In both the SNP and nicardipine groups, the arteriole diameter was increased after the assigned treatment and remained above the pre-arrest baseline value. However, the increase in the arteriole diameter was greater in the SNP group. The mixed-effects model for the %arteriole diameter during the first 5 min showed a significant group-time interaction among the three groups ($P = 0.035$). The model revealed a significant increase over time in both the nicardipine (coefficient [95% CI], 7.978 [0.329–15.627]) and SNP (coefficient [95% CI], 9.373 [1.724–17.022]) groups compared to the saline group. The mixed-effects model for the %arteriole diameter from 5 min to 5 h showed a significant group effect ($P = 0.022$). In the model, the %arteriole diameter was higher in the SNP group (coefficient [95% CI], 48.970 [13.884–84.057]) but not in the nicardipine group compared to the saline group. The MFI of the SNP group was steadily increased after the assigned treatment, whereas the MFI of the nicardipine group was almost identical to that of the saline group. The mixed-effects model for the MFI from 5 min to 5 h showed a significant group effect ($P = 0.024$). In the model, the MFI was higher in the SNP group (coefficient [95% CI], 0.296 [0.071–0.521]) but not in the nicardipine group compared to the saline group.

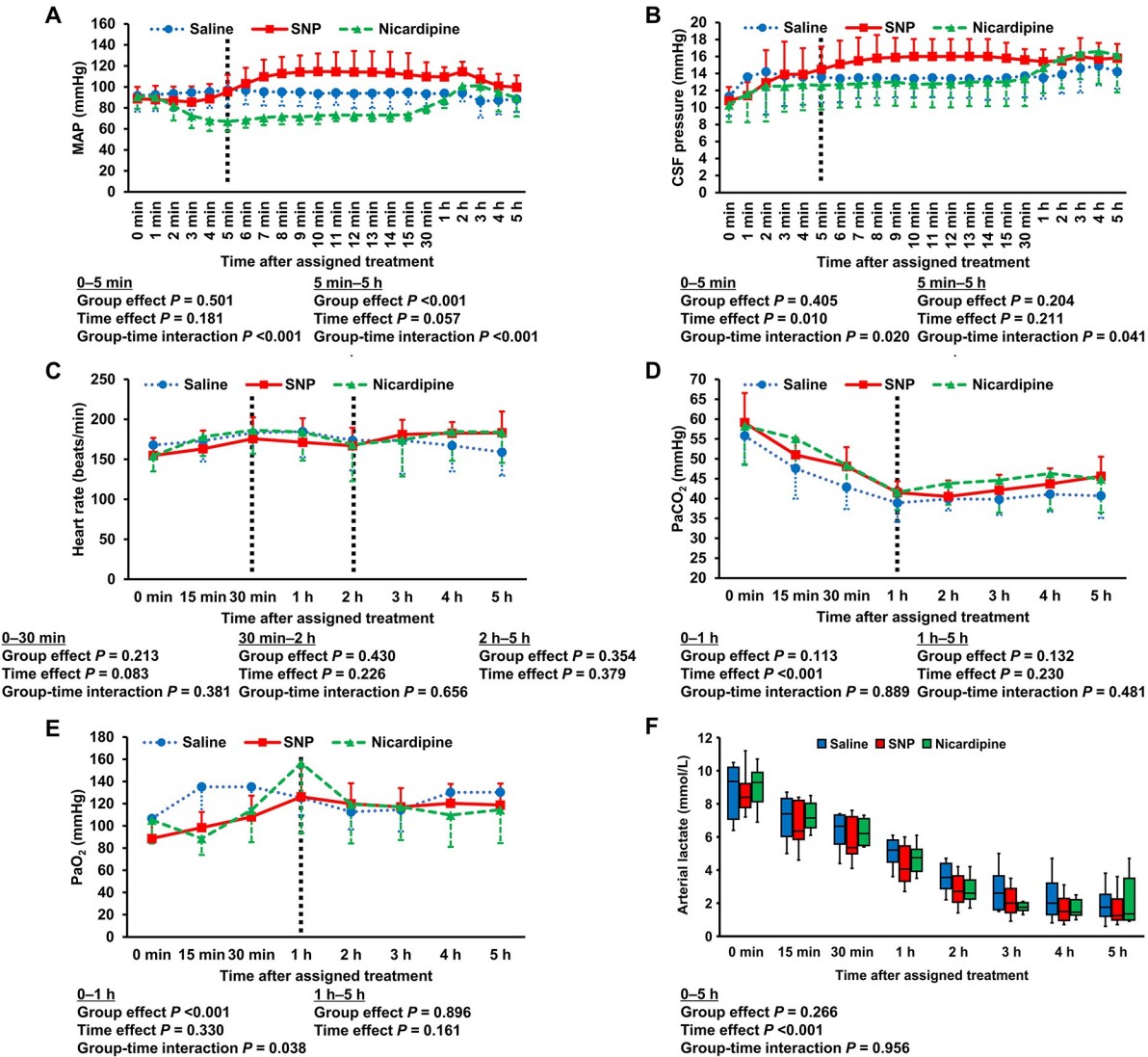

**Fig 2.** MAP (A), CSF pressure (B), heart rate (C), PaCO$_2$ (D), PaO$_2$ (E), and arterial lactate (F) after the assigned treatment. Data are presented as mean ± SD or box-whisker plots. *P* values were derived from linear mixed-effects models. SNP, sodium nitroprusside; MAP, mean arterial pressure; CSF, cerebrospinal fluid; PaCO$_2$, partial pressure of arterial carbon dioxide; PaO$_2$, partial pressure of arterial oxygen.

Fig 4 shows the PctO$_2$ and mean aEEG amplitude after the assigned treatment. In the saline group, PctO$_2$ was gradually decreased, remaining below the pre-arrest baseline levels for approximately 2 h from 15 min after the assigned treatment, followed by a slow increase to approximately 58 mmHg. SNP administration immediately induced a marked increase in PctO$_2$. In the SNP group, PctO$_2$ was slowly decreased after 15 min following SNP administration but remained above the pre-arrest baseline levels until the end of the post-ROSC intensive care period. Nicardipine administration induced a gradual increase in PctO$_2$ over 4 h without an initial rapid increase as observed after SNP administration. The mixed-effects model for PctO$_2$ in the 5 h period showed a significant group effect (*P* <0.001) and group-time interaction (*P* <0.001). In the model, PctO$_2$ was higher in the SNP group (coefficient [95% CI], 26.926 [12.404–41.449]) but not in the nicardipine group compared to the saline group. In all animals, the mean aEEG amplitude was markedly depressed immediately prior to the assigned

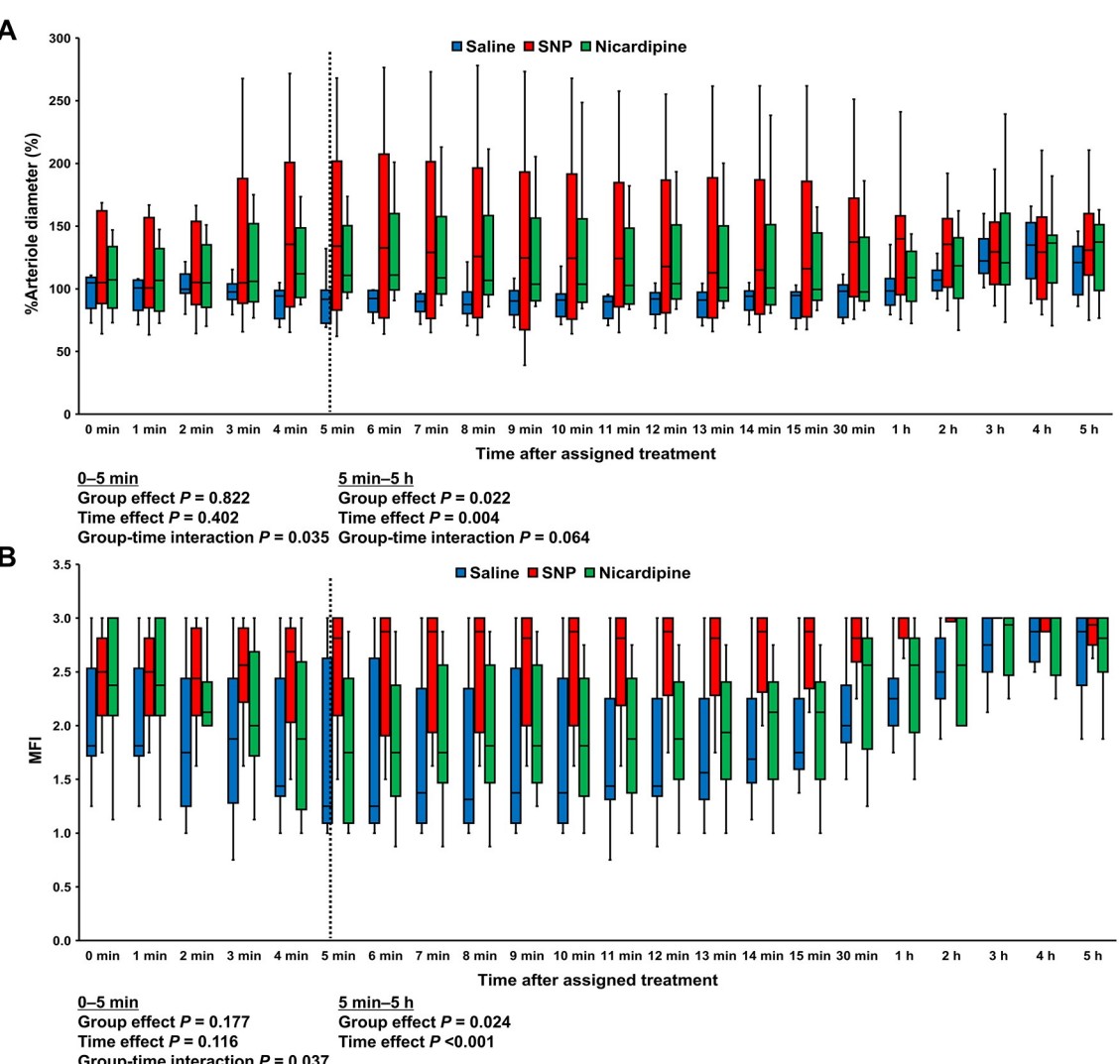

**Fig 3.** %Arteriole diameter (A) and MFI (B) after the assigned treatment. Data are presented as box-whisker plots. *P* values were derived from linear mixed-effects models. SNP, sodium nitroprusside; MFI, microvascular flow index.

treatment and gradually recovered over time. The mixed-effects model for the mean aEEG amplitude in the 5 h period demonstrated a significant group-time interaction among the three groups ($P < 0.001$). The mean aEEG amplitude showed a faster recovery in the SNP group (coefficient [95% CI], 1.149 [0.468–1.829]) but not in the nicardipine group compared to the saline group.

## Discussion

In the present study, a significant decrease in the MAP was observed after nicardipine administration, whereas no such decrease was observed after SNP administration. In contrast to nicardipine administration, SNP administration significantly increased the pial arteriolar diameter, MFI, and PctO$_2$ and promoted aEEG amplitude recovery. These findings suggest that SNP is more suitable than nicardipine for the treatment of post-ischemic cerebral hypoperfusion following cardiac arrest.

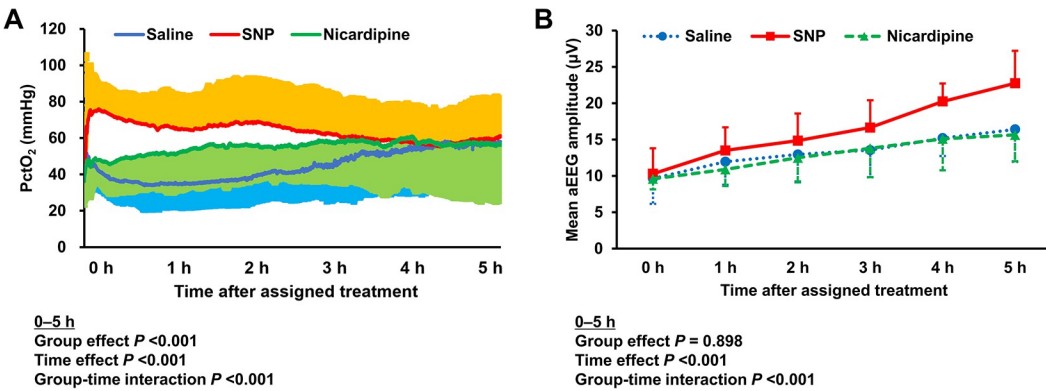

**Fig 4.** PctO₂ (A) and mean aEEG amplitude (B) after the assigned treatment. Data are presented as mean ± SD. *P* values were derived from linear mixed-effects models. SNP, sodium nitroprusside; PctO₂, cortical tissue oxygen tension; aEEG, amplitude-integrated electroencephalography.

In previous studies [12,13], post-ischemic cerebral hypoperfusion was successfully reversed after intrathecal vasodilator administration, suggesting that post-ischemic cerebral hypoperfusion may be caused by cerebral vasospasm due to an imbalance between local vasoconstrictive and vasodilatory mediators rather than structural derangement of the vascular wall, such as edema and intimal hyperplasia [6,25,26]. A recent study indicated that constriction of pial arterioles on the surface of the brain is responsible for post-ischemic cerebral hypoperfusion after cardiac arrest [27]. Because of the dissociation between cerebral microcirculation and macrovascular circulation [28–30], vasodilator administration rather than blood pressure manipulation may be required to sufficiently improve CBF in the early post-ROSC period. Following systemic administration, a vasodilator drug may not reach the vasospastic pial arterioles in a sufficient concentration to reverse vasospasm unless doses sufficient to cause severe hypotension are used [12]. In contrast, intrathecal administration into the subarachnoid space allows direct delivery of a sufficiently high concentration of a vasodilator to the vasospastic pial arterioles while minimizing severe side effects associated with systemic administration, such as hypotension [12,31], making it an attractive option for the treatment of post-ischemic cerebral hypoperfusion following cardiac arrest.

To the best of our knowledge, this is the first study comparing the efficacy of intrathecal SNP and nicardipine in treating post-ischemic cerebral hypoperfusion following cardiac arrest. Previous studies compared the efficacy of intravenous administrations of these two vasodilators in various clinical settings and reported their comparable blood pressure-lowering efficacy [32,33]. In a study by Matot et al. comparing pulmonary vasodilator responses to intralobar injections of several vasodilators in cats [34], there was no significant difference between the pulmonary vasodilator response to SNP and that to nicardipine. They also reported that the doses of SNP and nicardipine required for a 10 mmHg reduction in lobar arterial pressure were comparable (55 ± 15 μg and 44 ± 16 μg, respectively). The reason for the ineffectiveness of nicardipine in treating post-ischemic cerebral hypoperfusion in our study is unclear but may be related to MAP reduction following nicardipine administration. Changes in cerebral perfusion after vasodilator treatment depend not only on the degree of cerebral vasodilation but also on the cerebral perfusion pressure, which is mainly dependent on the MAP. The decrease in the MAP after nicardipine administration could have markedly attenuated the CBF-promoting effect of intrathecally administered nicardipine. The significant decrease in the MAP following intrathecal nicardipine administration in this study suggests that a significant amount of nicardipine might have been redistributed to systemic circulation. Voldby

et al. reported that nimodipine, another calcium channel blocker with pharmacological properties similar to those of nicardipine, was detected in the blood within 5 min after intrathecal administration [31]. A decrease in nicardipine concentration in the CSF due to redistribution to systemic circulation could have attenuated pial arteriolar dilation by nicardipine.

Interestingly, a transient small decrease in the MAP followed by an increase in the MAP was observed after intrathecal SNP administration in the present study. Similar biphasic pressor responses to intrathecal SNP administration have been reported in previous studies [35–37]. Garcia et al. suggested that the initial vasodepressor response and subsequent pressor response elicited by intrathecal SNP injection could be attributed to nitric oxide-induced release of γ-aminobutyric acid and glutamate, respectively, in the spinal cord [36]. Further studies are required to clarify the hemodynamic effects of intrathecally administered SNP.

Intravenously administered SNP is known to have a very short action duration of several minutes. However, consistent with previous studies [12,13], the effect of a single intrathecal injection of SNP seemed sufficient (approximately 2 h) in the present study, eliminating the need for repeated intrathecal SNP injections. The prolonged action might be explained by the ability of intrathecally administered SNP to bypass systemic metabolic processes, thereby maintaining therapeutic drug concentrations in the CSF space for an extended period of time. Post-ischemic cerebral hypoperfusion was improved within 3 h after ROSC in the saline group in the present study; however, it may persist much longer [38]. In this case, post-ischemic cerebral hypoperfusion may recur if the SNP concentration in the CSF decreases below the therapeutic range, making repeated intrathecal SNP injections necessary. Further studies are needed to determine whether a single intrathecal SNP injection can adequately treat post-ischemic cerebral hypoperfusion following cardiac arrest.

One of the major concerns with intrathecal vasodilator administration is the potential risk of increased intracranial pressure. Cerebral vasodilation caused by intrathecal vasodilator administration may significantly increase intracranial pressure through increased cerebral blood volume. A small increase in CSF pressure, which is thought to be caused by a mass effect due to the volume of saline injected into the subarachnoid space, was observed immediately following intrathecal administration in all animals in the present study. However, no significant difference in CSF pressure was observed among the three groups, suggesting that intrathecally administered SNP and nicardipine themselves did not have a significant effect. The small increase in CSF pressure immediately following intrathecal administration may be prevented by withdrawing a volume of CSF equal to the volume to be administered prior to intrathecal administration.

Our study has several limitations. First, this study was conducted using anesthetized young pigs without underlying diseases. Although pigs have a cerebrovascular anatomy similar to that of humans, there are differences in CSF volume and flow dynamics [39]. The use of sevoflurane anesthesia may have affected hemodynamic responses to intrathecal vasodilator therapy as well as cerebral measurements in the present study [40,41]. Therefore, the results of our study may not be readily generalized to human patients with cardiac arrest. Second, we did not optimize the doses of SNP and nicardipine. We administered SNP and nicardipine at doses consistent with those used in clinical studies [10,14,23]. Using the animal equivalent dose (0.18 mg/kg) obtained based on body surface area normalization, we calculated the doses to be administered to our animals as 4.17 mg. This dose is close to the dose of 4 mg used in the clinical studies [10,14,23]. Given the larger volume and faster circulation of CSF in adult humans compared to young pigs [39,42], the CSF drug concentrations and their CBF-promoting effects after intrathecal administration of the same doses may be smaller in adult patients with cardiac arrest than in the animals in this study. Third, our study was performed using a model with ischemic insult caused by a single duration of untreated ventricular fibrillation.

The severity and temporal pattern of post-ischemic cerebral hypoperfusion following cardiac arrest are dependent on the type of cardiac arrest (ventricular fibrillation versus asphyxia) and duration of cardiac arrest [43,44]. Our findings need to be verified in models with different cardiac arrest characteristics. Fourth, we did not examine serum neurobiomarkers, nor did we assess functional neurologic or histologic outcomes. Longer-term care and observation of the animals are necessary to accurately assess these parameters, as neurologic consequences of HIBI are established over several days [45,46]. However, we were unable to support and house the animals for a long duration because of our limited resources. A study to determine the effects of intrathecal SNP administration on functional and histologic outcomes after cardiac arrest is currently underway in our laboratory. Fifth, we could not determine the impact of intrathecal vasodilator administration on penetrating vessels, capillaries, or large cerebral arteries, as we observed only pial arterioles on the surface of the brain. To better understand the effects of intrathecal vasodilator administration on the cerebrovascular system, additional studies using tools such as multiphoton microscopy and transcranial Doppler are needed. Sixth, only SNP and nicardipine were tested in this study. Other intrathecal vasodilators need to be tested to identify optimal vasodilators for treating post-ischemic cerebral hypoperfusion following cardiac arrest.

## Conclusions

In the present study that examined the effects of intrathecal administration of SNP and nicardipine on pial microcirculation, cortical tissue oxygenation, and electrocortical activity in pigs resuscitated from cardiac arrest, intrathecal SNP but not nicardipine was effective in treating post-ischemic cerebral hypoperfusion after cardiac arrest.

## Supporting information

**S1 Data. Raw data.**
(XLSX)

**S1 File. ARRIVE guidelines checklist.**
(DOC)

## Author Contributions

**Conceptualization:** Kyung Woon Jeung.

**Formal analysis:** Wan Young Heo, Yong Soo Cho, Byung Kook Lee.

**Funding acquisition:** Kyung Woon Jeung.

**Investigation:** Hyoung Youn Lee, Najmiddin Mamadjonov, Kyung Woon Jeung, Yong Hun Jung.

**Methodology:** Hyoung Youn Lee, Najmiddin Mamadjonov, Kyung Woon Jeung, Yong Hun Jung.

**Resources:** Yong Hun Jung.

**Supervision:** Tag Heo.

**Writing – original draft:** Hyoung Youn Lee, Najmiddin Mamadjonov, Kyung Woon Jeung, Yong Hun Jung, Wan Young Heo, Yong Soo Cho, Byung Kook Lee, Tag Heo.

**Writing – review & editing:** Kyung Woon Jeung, Tag Heo.

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
