## [Decision Letter · Decision Letter 0]

4 Sep 2024

PONE-D-24-30125Effects of intrathecal administration of sodium nitroprusside and nicardipine on cerebral pial microcirculation, cortical tissue oxygenation, and electrocortical activity in the early post-resuscitation period in a porcine cardiac arrest modelPLOS ONE

Dear Dr. Jeung,

Thank you for submitting your manuscript to PLOS ONE. After careful consideration, we feel that it has merit but does not fully meet PLOS ONE’s publication criteria as it currently stands. Therefore, we invite you to submit a revised version of the manuscript that addresses the points raised during the review process.

The manuscript is well evaluated by the reviewer.Respond to the reviewer's comments appropriately.

We look forward to receiving your revised manuscript.

Kind regards,

Masaki Mogi

Academic Editor

PLOS ONE

Journal Requirements:

1. When submitting your revision, we need you to address these additional requirements. Please ensure that your manuscript meets PLOS ONE's style requirements, including those for file naming. The PLOS ONE style templates can be found at https://journals.plos.org/plosone/s/file?id=wjVg/PLOSOne_formatting_sample_main_body.pdf and https://journals.plos.org/plosone/s/file?id=ba62/PLOSOne_formatting_sample_title_authors_affiliations.pdf 2. To comply with PLOS ONE submissions requirements, in your Methods section, please provide additional information regarding the experiments involving animals and ensure you have included details on methods of sacrifice. 3. We note that there is identifying data in the Supporting Information file <S1_Data>. Due to the inclusion of these potentially identifying data, we have removed this file from your file inventory. Prior to sharing human research participant data, authors should consult with an ethics committee to ensure data are shared in accordance with participant consent and all applicable local laws. Data sharing should never compromise participant privacy. It is therefore not appropriate to publicly share personally identifiable data on human research participants. The following are examples of data that should not be shared: -Name, initials, physical address-Ages more specific than whole numbers-Internet protocol (IP) address-Specific dates (birth dates, death dates, examination dates, etc.)-Contact information such as phone number or email address-Location data-ID numbers that seem specific (long numbers, include initials, titled “Hospital ID”) rather than random (small numbers in numerical order) Data that are not directly identifying may also be inappropriate to share, as in combination they can become identifying. For example, data collected from a small group of participants, vulnerable populations, or private groups should not be shared if they involve indirect identifiers (such as sex, ethnicity, location, etc.) that may risk the identification of study participants. Additional guidance on preparing raw data for publication can be found in our Data Policy (https://journals.plos.org/plosone/s/data-availability#loc-human-research-participant-data-and-other-sensitive-data) and in the following article: http://www.bmj.com/content/340/bmj.c181.long. Please remove or anonymize all personal information (<specific identifying information in file to be removed>), ensure that the data shared are in accordance with participant consent, and re-upload a fully anonymized data set. Please note that spreadsheet columns with personal information must be removed and not hidden as all hidden columns will appear in the published file.

Reviewers' comments:

Reviewer's Responses to Questions

**Comments to the Author**

1. Is the manuscript technically sound, and do the data support the conclusions?

Reviewer #1: Yes

2. Has the statistical analysis been performed appropriately and rigorously? 

Reviewer #1: Yes

3. Have the authors made all data underlying the findings in their manuscript fully available?

Reviewer #1: Yes

4. Is the manuscript presented in an intelligible fashion and written in standard English?

Reviewer #1: Yes

5. Review Comments to the Author

Reviewer #1: The reviewed manuscript written by Hyoung Youn Lee and co-authors aims to assess the effects of intrathecal administration of sodium nitroprusside and nicardipine on various brain tissue parameters. These include cerebral pial microcirculation, cortical tissue oxygenation, and electrocortical activity in a porcine cardiac arrest model after successful resuscitation.

The manuscript is well-written and structured, the presented data and appropriate statistical analysis support the Author's arguments and conclusions. The study was carried out with all appropriate ethical aspects, and I have just few minor notes to address:

1. I think that the Authors should extend and highlight the Limitations section at the end of the discussion by adding the information about other methods that might have been included in this study in order to better evaluate the effects in a comprehensive manner. For example, they can mention the other methods that make a real-time assessment of cerebral blood flow (LSCI, Photoplethysmography, Doppler, etc.) or brain tissue biopsy/autopsy to assess structural effects on the cerebral cortex tissue. This can show the perspectives for further investigations in this direction.

2. The Authors mention that the doses of sodium nitroprusside and nicardipine were chosen based on clinical studies, have the Authors assessed these doses using the dose conversion between animals and human via equivalent dose calculation? How does it correspond?

6. PLOS authors have the option to publish the peer review history of their article (what does this mean?). If published, this will include your full peer review and any attached files.

Reviewer #1: **Yes: **Gennadii Piavchenko

---

## [Author Response · Author response to Decision Letter 0]

16 Oct 2024

Dear Editor and Reviewer #1,

Firstly, we appreciate you for your comments. They were very helpful in improving our manuscript and included very useful points that we had not previously recognized. After due consideration, the manuscript was revised as described below.

Journal Requirements:

: Our manuscript was revised to meet PLOS ONE’s style requirements.

2. To comply with PLOS ONE submissions requirements, in your Methods section, please provide additional information regarding the experiments involving animals and ensure you have included details on methods of sacrifice.

: The following sentences are included in the method section to comply with PLOS ONE submissions requirements.

- All experiments were approved by the Animal Care and Use Committee of Chonnam National University Hospital (CNUH IACUC-22028) and performed in accordance with the National Institutes of Health Guide for the Care and Use of Laboratory Animals. The investigators who performed the experiments had completed an Institutional Animal Care and Use Committee training course on animal care and handling. All surgical interventions were performed under sevoflurane anesthesia, and every effort was made to prevent unnecessary suffering of the animals.

- After intramuscular administration of ketamine (20 mg/kg) and xylazine (2.2 mg/kg), they were anesthetized with sevoflurane, endotracheally intubated, and mechanically ventilated with a mixture of nitrous oxide and oxygen (7:3). Sevoflurane was titrated to achieve an appropriate anesthetic depth (no pedal withdrawal reflex and no increase in arterial pressure or heart rate).

- Animals that achieved ROSC received 5.5 h of intensive care under sevoflurane anesthesia.

- None of the animals reached the predetermined humane endpoints for euthanasia (systolic arterial pressure <60 mmHg, heart rate <40 beats/min, or seizure) until the end of the post-ROSC intensive care period.

- Immediately after the post-ROSC intensive care period, the animals were humanely euthanized with a rapid intravenous bolus injection of potassium chloride under deep sevoflurane anesthesia.

In addition, the title and abstract indicate that this is an animal experiment. The ARRIVE guidelines checklist is submitted (S1 File).

3. We note that there is identifying data in the Supporting Information file <S1_Data>. Due to the inclusion of these potentially identifying data, we have removed this file from your file inventory. Prior to sharing human research participant data, authors should consult with an ethics committee to ensure data are shared in accordance with participant consent and all applicable local laws.

-Location data

: All identifying data has been removed from the S1 Data file.

: We reviewed the reference list. We ensure that the reference list is complete and correct. The changes to the reference list were mentioned in the rebuttal letter that accompanies our revised manuscript.

Reviewers' comments:

Reviewer's Responses to Questions

Comments to the Author

1. Is the manuscript technically sound, and do the data support the conclusions?

Reviewer #1: Yes

2. Has the statistical analysis been performed appropriately and rigorously?

Reviewer #1: Yes

3. Have the authors made all data underlying the findings in their manuscript fully available?

Reviewer #1: Yes

4. Is the manuscript presented in an intelligible fashion and written in standard English?

Reviewer #1: Yes

5. Review Comments to the Author

Reviewer #1: The reviewed manuscript written by Hyoung Youn Lee and co-authors aims to assess the effects of intrathecal administration of sodium nitroprusside and nicardipine on various brain tissue parameters. These include cerebral pial microcirculation, cortical tissue oxygenation, and electrocortical activity in a porcine cardiac arrest model after successful resuscitation.

The manuscript is well-written and structured, the presented data and appropriate statistical analysis support the Author's arguments and conclusions. The study was carried out with all appropriate ethical aspects, and I have just few minor notes to address:

1. I think that the Authors should extend and highlight the Limitations section at the end of the discussion by adding the information about other methods that might have been included in this study in order to better evaluate the effects in a comprehensive manner. For example, they can mention the other methods that make a real-time assessment of cerebral blood flow (LSCI, Photoplethysmography, Doppler, etc.) or brain tissue biopsy/autopsy to assess structural effects on the cerebral cortex tissue. This can show the perspectives for further investigations in this direction.

: We observed only pial arterioles on the surface of the brain among the cerebrovascular system. To address this limitation, the sentences “Fifth, we could not determine the impact of intrathecal vasodilator administration on penetrating vessels, capillaries, or large cerebral arteries, as we observed only pial arterioles on the surface of the brain. To better understand the effects of intrathecal vasodilator administration on the cerebrovascular system, additional studies using tools such as multiphoton microscopy and transcranial Doppler are needed.” were added to the limitation section.

We did not examine serum biomarkers of brain injury, nor did we assess functional neurologic or histologic outcome. These assessments require support and observation of the animals for a prolonged period of time, as biochemical, histologic, and functional neurologic consequences of hypoxic-ischemic brain injury are established over several days. However, because of our limited resources (shortage of personnel), we could not support and keep the animals for a long time period. As you pointed out above, biomarkers of brain injury and/or histologic findings, even when assessed in the early period after ROSC, might have provided important clues about the effects of intrathecal vasodilator administration.

To address this issue, the sentences “Third, we did not determine long-term neurologic outcomes. In the future, we plan to determine whether the CBF-promoting effect of intrathecal SNP administration is beneficial for long-term neurologic outcomes.” in the discussion section were changed to “Fourth, we did not examine serum neurobiomarkers, nor did we assess functional neurologic or histologic outcomes. Longer-term care and observation of the animals are necessary to accurately assess these parameters, as neurologic consequences of HIBI are established over several days [45,46]. However, we were unable to support and house the animals for a long duration because of our limited resources. A study to determine the effects of intrathecal SNP administration on functional and histologic outcomes after cardiac arrest is currently underway in our laboratory.”.

Accordingly, the following two references were added to the reference list:

45. Vondrakova D, Kruger A, Janotka M, Malek F, Dudkova V, Neuzil P, et al. Association of neuron-specific enolase values with outcomes in cardiac arrest survivors is dependent on the time of sample collection. Crit Care. 2017;21: 172.

46. Back T, Schüler OG. The natural course of lesion development in brain ischemia. Acta Neurochir Suppl. 2004;89: 55–61.

2. The Authors mention that the doses of sodium nitroprusside and nicardipine were chosen based on clinical studies, have the Authors assessed these doses using the dose conversion between animals and human via equivalent dose calculation? How does it correspond?

: We administered SNP and nicardipine at doses consistent with those used in clinical studies (Thomas et al. Stroke. 1999;30: 1409–1416; Pathak et al. Br J Neurosurg. 2003;17: 306–310; Hafeez et al. Neurocrit Care. 2019;31: 399–405). Using the animal equivalent dose (0.18 mg/kg) obtained based on body surface area normalization, we calculated the doses to be administered to our animals as 4.17 mg. This dose is close to the dose of 4 mg used in the clinical studies. Given the larger volume and faster circulation of CSF in adult humans compared to young pigs, the CSF drug concentrations and their CBF-promoting effects after intrathecal administration of the same doses may be smaller in adult patients with cardiac arrest than in the animals in this study.

To address issue, the sentences “Second, we did not optimize the doses of SNP and nicardipine. We administered SNP and nicardipine at doses consistent with those used in clinical studies [10,14,23]. Using the animal equivalent dose (0.18 mg/kg) obtained based on body surface area normalization, we calculated the doses to be administered to our animals as 4.17 mg. This dose is close to the dose of 4 mg used in the clinical studies [10,14,23]. Given the larger volume and faster circulation of CSF in adult humans compared to young pigs [39,42], the CSF drug concentrations and their CBF-promoting effects after intrathecal administration of the same doses may be smaller in adult patients with cardiac arrest than in the animals in this study.” were added to the limitation section.

Accordingly, the following reference was added to the reference list:

42. Fil JE, Joung S, Zimmerman BJ, Sutton BP, Dilger RN. High-resolution magnetic resonance imaging-based atlases for the young and adolescent domesticated pig (Sus scrofa). J Neurosci Methods. 2021;354: 109107.

Thank you again for your invaluable suggestions for improving our manuscript.

Sincerely,

---

## [Decision Letter · Decision Letter 1]

22 Oct 2024

Effects of intrathecal administration of sodium nitroprusside and nicardipine on cerebral pial microcirculation, cortical tissue oxygenation, and electrocortical activity in the early post-resuscitation period in a porcine cardiac arrest model

PONE-D-24-30125R1

Dear Dr. Jeung,

We’re pleased to inform you that your manuscript has been judged scientifically suitable for publication and will be formally accepted for publication once it meets all outstanding technical requirements.

Kind regards,

Masaki Mogi

Academic Editor

PLOS ONE

Additional Editor Comments (optional):

Reviewers' comments:

Reviewer's Responses to Questions

**Comments to the Author**

1. If the authors have adequately addressed your comments raised in a previous round of review and you feel that this manuscript is now acceptable for publication, you may indicate that here to bypass the “Comments to the Author” section, enter your conflict of interest statement in the “Confidential to Editor” section, and submit your "Accept" recommendation.

Reviewer #1: All comments have been addressed

2. Is the manuscript technically sound, and do the data support the conclusions?

Reviewer #1: Yes

3. Has the statistical analysis been performed appropriately and rigorously? 

Reviewer #1: Yes

4. Have the authors made all data underlying the findings in their manuscript fully available?

Reviewer #1: Yes

5. Is the manuscript presented in an intelligible fashion and written in standard English?

Reviewer #1: Yes

6. Review Comments to the Author

Reviewer #1: The authors corrected the manuscript and now it may be accepted for the publication in PLOS One journal.

7. PLOS authors have the option to publish the peer review history of their article (what does this mean?). If published, this will include your full peer review and any attached files.

Reviewer #1: No

---

## [Editor Report · Acceptance letter]

24 Oct 2024

PONE-D-24-30125R1 

PLOS ONE

Dear Dr. Jeung, 

I'm pleased to inform you that your manuscript has been deemed suitable for publication in PLOS ONE. Congratulations! Your manuscript is now being handed over to our production team.

Kind regards, 

on behalf of

Dr. Masaki Mogi 

Academic Editor

PLOS ONE